# Impact of Internet Development on Carbon Emissions in Jiangsu, China

**DOI:** 10.3390/ijerph192416681

**Published:** 2022-12-12

**Authors:** Shijin Wang, Fan Tong

**Affiliations:** School of Business, Jiangsu Normal University, Xuzhou 221116, China

**Keywords:** carbon emissions, STIRPAT model, threshold effect, influence factors

## Abstract

Based on STIRPAT and panel threshold models, this study empirically tested the impact of Internet development on carbon emissions using panel data of Jiangsu Province from 2007 to 2020. The results showed that the carbon emissions intensity of the Internet development level had a significant promotion effect, while the carbon emissions intensity of technological progress showed a significant inhibition effect, but this inhibition effect is less than the promotion effect brought about by internet development. Considering the threshold effect, the development of the Internet had a double-threshold effect on carbon emissions in northern and central Jiangsu. Jiangsu Province should further accelerate the pace of Internet development and cross the threshold value as soon as possible. Finally, this study constructed a prediction model of emissions reduction to predict the future emissions reduction potential of Jiangsu Province and found that there was still much room for improvement regarding carbon emissions reduction in Jiangsu Province.

## 1. Introduction

In recent decades, the rapid development of the Internet has not only had a profound impact on people′s lives but also contributed to the economic development of various productive sectors in China; however, this economic development has also brought environmental problems with it. Given the background of climate warming, the development of a circular economy has become the consensus of the international community. China has made progress in economic development, but still faces tremendous pressure to reduce carbon emissions. According to statistics, China′s emissions are the sum of all developed countries, ranking first in the world. At present, the Internet in China is developing rapidly and has penetrated all walks of life. Therefore, it is of great significance to study whether and what impact the development of the Internet will have on carbon emissions.

Jiangsu Province is an important constituent province of the Yangtze River Economic Belt. Its comprehensive strength puts it in the leading position in the country. Not only is its economic development in good shape but its Internet development is also far ahead. However, there are differences in the development between different regions in the province. The economic differences between southern and northern Jiangsu are quite large. Given this background, this study selected Jiangsu Province with its unique geographical location and developed economy as the research object, which is of representative significance.

The novelty and marginal contribution of this study are as follows: this study (1) constructed an extended STIRPAT model that took into account factors such as Internet development and technological progress and analyzed the influencing factors of carbon emissions in Jiangsu Province, (2) analyzed the impact of the Internet on carbon emissions in Jiangsu Province through the threshold effect and (3) built a carbon emissions reduction prediction model to analyze the future carbon emission reduction potential of Jiangsu Province.

## 2. Literature Review

Most studies focused on the impact of economic growth, industrial structure, energy structure and other factors on carbon emissions. Population growth is one of the sources of carbon emissions, and decoupling population development from carbon emissions is of great significance in the realization of carbon emissions reduction [1]. Based on the extended STIRPAT model, the driving factors that affect the HCE in Jiangsu Province are household size, total population, unemployment rate, urbanization rate and aging, among which household size has the greatest effect [2]. As the size of the economy continues to expand and people′s lifestyles change dramatically, the role of demographic factors on carbon emissions is diminishing, but due to the limited level of technology, the value of carbon emissions reductions due to technological improvements is much smaller than the value of carbon emission increases due to lifestyle changes. The impact of various factors on carbon emissions is different in the region, such as economic development in the east and west, causing a shift in the center and intensity of carbon emissions to the west. High-energy-consuming industries are gradually gathering in the central and western regions, which has a positive spatial spillover effect on environmental pollution, and the long-term effect is greater than the short-term effect. This industrial transfer policy has reduced environmental pollution in the east, but the environmental and economic efficiency in the central and western regions has decreased [3]. Moreover, taking the Yangtze River Delta region as an example, the socio-economic and environmental sustainability have been improved, but there are huge differences between different regions, among which 78.79% of the counties improved their socio-economic sustainability, but their environmental sustainability declined in the same period [4]. With the accelerated pace of industrialization in China, the consumption of energy is also increasing. Oil consumption in the transportation sector accounts for 38.2% of the total oil demand, significantly increasing CO_2_ emissions [5]. In the case of Sichuan Province, due to the optimization of the energy structure, the use of clean energy is being paid more and more attention due to its emission reduction efficiency [6].

With the popularity of the Internet, there are also many scholars who have studied the impact of Internet development on carbon emissions; though the conclusions have not yet been unified, the following views have developed.

For one thing, with the development of the Internet, carbon emissions will increase. In the case of EU countries, Internet use, economic development and trade openness can have a positive impact on carbon emissions [7]. ICT and economic development can also contribute to an increase in carbon emissions, and after the introduction of the interaction between them, ICT and financial development play a moderating role in stimulating the growth of carbon emissions. Therefore, it is essential to promote green ICT development in the financial sector to improve energy efficiency [8]. 

Second, the development of the Internet has curbed carbon emissions. A study found that the Internet improves energy efficiency and enhances carbon emissions performance by promoting industrial structure upgrading and technological development to achieve green and sustainable economic development [9]. Decomposing DI on China′s implied carbon emissions into direct structural effects and indirect structural effects, results show that the direct structural effects of DI factors have a significant negative effect on China′s implied carbon emissions [10]. Relevant data on the digital economy and environmental pollution in China indicate that there is a spatial interaction spillover effect between the digital economy and urban environmental pollution in China, and the local digital economy and environmental pollution restrict each other [11]. The digital economy has a significant inhibitory effect on carbon emissions, and green efficiency can affect the degree of effect between the two [12].

In summary, the influencing factors of carbon emissions are mentioned in the current research, but there are still the following shortcomings: (1) the research on the influencing factors of carbon emissions focused on the energy structure, industrial structure and economic development, and (2) the research direction of Internet development mainly focused on the impact of Internet development on the economy. There is little research on the relationship between Internet development and carbon emissions intensity, and the research results in this area are controversial. This study used data from 13 cities in Jiangsu Province for the empirical analysis. First, we comparatively analyzed the influencing factors of carbon strength in the context of Internet development in various regions of the province. Second, the potential for future abatement in Jiangsu Province was analyzed. This study can not only supplement and enrich the research foundation of the mechanism of the Internet development on carbon emissions, but also provide a basis for the development of the Internet and environmental protection in Jiangsu, and has certain reference value for the development of other provinces.

## 3. Research Design 

### 3.1. Carbon Intensity Measurement

Carbon intensity is the carbon dioxide emissions per unit of GDP. This study calculated the total carbon emissions based on the total energy consumption. The calculation formula is
(1)$$C=\sum_{j=1}^{8}E_{j}*F_{j}$$

In Formula (1), *C* is the carbon emissions; *E_j_* is the consumption of energy source *j*, where *j* = 1,2,3…8; and *F_j_* is the carbon emissions coefficient of *j*, which is shown in Table 1.

Calculated from Equation (1), the carbon emissions of Jiangsu and each region are shown in Figure 1. Overall, the carbon emissions of Jiangsu Province and various regions showed an upward trend. The total carbon emissions of Jiangsu Province increased from 139.189 million tons to 47.204 million tons from 2007 to 2020, with an average annual growth rate of 32.39%. From a regional perspective, the growth rate of carbon emissions in Southern Jiangsu was the fastest, from 111.4734 million tons to 292.318 million tons from 2007 to 2020, with an average annual growth rate of 41.07%. The growth rate of carbon emissions in northern Jiangsu was second, from 16,248,100 tons in 2007 to 101,074,700 tons in 2020, an increase of 5.22 times, with an average annual growth rate of 18.50%. Compared with the southern and northern regions of Jiangsu, carbon emissions in the central region of Jiangsu grew at a slower rate, with an average annual growth rate of 16.91%. 

From the carbon intensity data (shown in Figure 2), it can be seen that the carbon intensity of Jiangsu Province decreased from 0.64 to 0.47 during 2007–2020, with an average carbon emissions intensity of 0.44. From 2007 to 2017, the carbon emissions intensity of northern, central and southern Jiangsu showed a downward trend as a whole. After 2017, the carbon emissions intensity of Jiangsu Province showed an upward trend. In the same year, the energy consumption intensity and economic level of each region in the province also obviously improved. The carbon emissions intensity in southern Jiangsu was higher than the average value of Jiangsu province, while northern and central Jiangsu were lower than the provincial average value, where the rapid economic development in southern Jiangsu opened a larger gap with the economies of central and northern Jiangsu. The development of secondary industry contributed the most to the economic growth in southern Jiangsu, but behind the rapid economic growth was a large amount of energy consumption, which led to a higher intensity of carbon emissions. In contrast, although the carbon emission intensity was lower in the central and northern Jiangsu Province, there was still much room for economic development, especially in the northern Jiangsu region, where the proportion of the secondary industry was only 41.9% on average, which was much lower than that in the southern and central Jiangsu regions. In a word, the coordination between economic development and environmental protection in Jiangsu Province has not yet been reached, and the pressure on emissions reduction was relatively high. 

### 3.2. Models 

Carbon emissions are mainly derived from the consumption of fossil energy, such as coal, oil and liquefied petroleum gas, and the main factors that affect the consumption of these fossil fuels are industrial structure, population size, per capita income, energy structure, economic development, etc. In the choice of analysis methods, most studies used the Kuznets curve, two-stage LMDI model, STIRPAT model, IPAT model and other methods to carry out factor analysis. For example, Tzeremes and Panayiotis (2019) confirmed the existence of the environmental Kuznets curve in various regions of China [13]; Peng (2022) established a two-stage LMDI model to study the influencing factors of carbon dioxide emissions from the fossil industry [14]. In addition, D. Liu (2018) analyzed the driving factors that affect CO_2_ emissions in China through the STIRPAT model [15]. According to the purpose of this study, we used the STIRPAT model to analyze the influencing factors of carbon intensity in Jiangsu. 

Dietz et al. (1994) proposed a stochastic environmental impact assessment model, namely, the STIRPAT model, based on the IPAT model [16]. The IPAT model is used to assess the impact of human behavior on the environment, where the expression is
(2)$$I_{t}{=\mathrm{aP}}^{b}{}_{t}A^{c}{}_{t}T^{d}{}_{t}\varepsilon_{t}$$

In Formula (2), I is the impact of human beings on the natural environment, which is usually expressed as the anthropogenic emissions of pollutants; P, A and T are independent variables that respectively represent the population size, wealth and technological level; a is the coefficient of the model; b, c, and d are the elasticity coefficients corresponding to P, A, and T, respectively, indicating that every time P, A and T change by 1%, changes of b%, c% and d% of I will be caused, respectively; t is the corresponding year; and ε is the random error item. 

In practice, in order to overcome the heteroskedasticity of the STIRPAT model, Formula (2) is usually converted into a logarithmic form: (3)$${\mathrm{lnI}}_{t}{\;=\;a\;+\;\mathrm{blnP}\;}_{t}{\;+\;\mathrm{clnA}}_{\;t}{\;+\;\mathrm{dlnT}}_{t}{\;+\;\varepsilon }_{t}\;$$

Based on the purpose of this study and the need for factor analysis, the traditional STIRPAT model is extended and converted into a logarithmic form:(4)$${\mathrm{lnI}}_{t}{\;=\;a}_{0}{\;+\;a}_{1}{\mathrm{lnP}}_{t}{\;+\;a}_{2}{\mathrm{lnA}}_{{}^{t}}{\;+\;a}_{3}{\mathrm{lnE}}_{t}{\;+\;a}_{4}{\mathrm{lnU}}_{t}{\;+\;a}_{5}{\mathrm{lnS}}_{t}{\;+\;a}_{6}{\mathrm{lnT}}_{t}{\;+\;a}_{7}{\mathrm{lnN}}_{t}{\;+\;\varepsilon }_{t\;}$$
where C represents the carbon emissions, t represents the corresponding year, the a_i_ values are constants and other variables are described in Table 2.

Since technological progress affects the level of Internet development to some extent, this study constructed a cross-product term between the level of Internet development and technological progress to construct model (5):(5)$${\mathrm{lnI}}_{t}{\;=\;a}_{0}{\;+\;a}_{1}{\mathrm{lnP}}_{t}{\;+\;a}_{2}{\mathrm{lnA}}_{t}{\;+\;a}_{3}{\mathrm{lnE}}_{t}{\;+\;a}_{4}{\mathrm{lnU}}_{t}{\;+\;a}_{5}{\mathrm{lnS}}_{t}{\;+\;a}_{6}{\mathrm{lnT}}_{t}{\;+\;a}_{7}{\mathrm{lnN}}_{t}{\;+\;a}_{8}({\mathrm{lnT}}_{t}\times {\mathrm{lnN}}_{t}){\;+\;\varepsilon }_{t}$$

In Formula (5), lnT_t×_lnN_t_ is the cross-product of the development level of the Internet and the technological progress, and the meanings of other letters are the same as in Formula (3).

In order to further study the relationship between Internet development and carbon emissions, the following single-panel threshold regression model was constructed:(6)$${\mathrm{lnI}}_{t}{\;=\;a}_{0}{+a}_{1}{\mathrm{lnP}}_{t}{\;+\;a}_{2}{\mathrm{lnA}}_{t}{\;+\;a}_{3}{\mathrm{lnE}}_{t}{\;+\;a}_{4}{\mathrm{lnU}}_{t}{\;+\;a}_{5}{\mathrm{lnS}}_{t}{\;+\;a}_{6}{\mathrm{lnT}}_{t}^{}{\;+\;a}_{7}{\mathrm{lnN}}_{t}I(q_{t}\leq \gamma_{1}){\;+\;a}_{8}{\mathrm{lnN}}_{t}I(q_{t}>\gamma_{1}){\;+\;\varepsilon }_{t}$$

The data for each variable were obtained from the 2007–2020 China City Statistical Yearbook, the China Industrial Statistical Yearbook and the city statistical yearbooks of each city in Jiangsu. The descriptive statistics of the main variables are shown in Table 3.

## 4. Results and Discussion

In order to prevent pseudo-regression, the stationarity of the time series of all variables needed to be tested before the model test. For this reason, the LLC test was selected to conduct a unit root test for each variable. If there is a unit root, it is a non-stationary time series [17]. The test results are shown in Table 4. It shows that all *p*-values were less than 0.05, the original hypothesis was rejected and all variables were stationary; therefore, the original equation could be regressed on this basis, but the type of panel model effect needed to be further determined [18].

The Hausman test is a common method to test whether fixed effect or random effect is used in the panel model [19]. Using Stata/SE16.0 to carry out the Hausman test on panel data, it can be seen that the Prob > chi^2^ value was 0.0383 from the test results, the original hypothesis was rejected at the 5% significance level and the fixed-effects model should be selected. The test results are shown in Table 5.

The regression results were obtained using Stata/SE16.0 software and are shown in Table 6, where model I is the regression result of the benchmark model and model II is the effect of considering the interaction effect of Internet development level and technological progress on the carbon intensity. As can be seen from model I in Table 6, the level of Internet development showed a significant contribution to the carbon intensity in Jiangsu, and the driving effects on carbon emissions in northern, central and southern Jiangsu were all significant at the 1% level, with coefficients of 0.234, 0.838 and 0.622, respectively. For Jiangsu as a whole, the inhibitory effect of technological development on carbon intensity was significant at the 1% level, and it also showed different degrees of inhibitory effect in the north, the center and the south. In addition, the inhibitory effect of technological development on carbon emissions was greater in southern Jiangsu than in northern Jiangsu and central Jiangsu. The coefficients of the industrial scale and urbanization level on the carbon intensity of Jiangsu province were both negative, but from the perspective of each region in the province, the industrial scale still had a driving effect on the carbon emissions in southern Jiangsu. The urbanization level had a suppressive effect on northern, central and southern Jiangsu, and the urbanization level in southern Jiangsu had the most significant impact on the carbon emissions. The level of economic development still showed a driving effect on the carbon emissions in Jiangsu, especially the economic development in the north, which had a coefficient of 0.678 on carbon emissions, while the economic development in the southern region showed a certain degree of inhibition on the carbon emissions. The driving effect on the carbon intensity in the north was significantly higher than in the center and south. 

From model II in Table 6, the coefficients of the interaction term between the level of Internet development and technological progress were 0.01, −0.006, 0.06 and 0.124 in Jiangsu overall, as well as in northern, central and southern Jiangsu respectively. With the introduction of the interaction term, the driving effect of the Internet development in Jiangsu on carbon intensity improved and southern Jiangsu showed a trend of suppression. However, in northern Jiangsu, carbon emissions were still at a high level as the level of Internet development increased. Technological progress could reduce carbon emissions caused by the development of the Internet, but the reduction was smaller than the increase caused by the development of the Internet [20].

Taking the level of Internet development as the threshold variable, the results of the threshold effect are shown in Table 7 and Table 8. Northern Jiangsu and central Jiangsu passed the single-threshold significance test with thresholds of −1.5360 and −1.1455, respectively.

As can be seen from Table 8, for the northern Jiangsu region, the coefficient of the inhibitory effect of the Internet development level on carbon emissions was −0.004 when lnN ≤ −1.5360, and when lnN > −1.5360, the Internet development showed a significant inhibitory effect at the 1% level with a coefficient of −0.209; it seems that Internet development’s ability to inhibit the carbon intensity increased by 0.209 percentage points for every 1% increase in the development level of the Internet. For the central Jiangsu region, when lnN ≤ −1.1415, the development level of the Internet increased the intensity of the carbon emissions at the 1% level with a coefficient of 0.438, while when lnN > −1.1415, the coefficient of the development level of the Internet on the intensity of carbon was −0.217. It can be seen that when the development level of the Internet crossed this threshold, the inhibitory effect of the development of the Internet on carbon emissions became greater, and there was variability in the effect of Internet development on the carbon intensity among the regions in Jiangsu [21].

According to the economies of scale, when an economy becomes larger and larger, it needs to consume more energy, and the growth of the economy will lead to more energy consumption, thus driving the carbon intensity higher [22]. However, when economic growth reaches a certain stage, it will lead to technological changes and the evolution of economic structure, which will lead to economic development that may lower the carbon intensity in a certain period [23]. The economic development of Jiangsu promoted an increase in carbon emissions. In addition, the economic level of Jiangsu showed a steady trend from 2007 to 2020, indicating that the economies of scale brought about by economic development were larger than the technology and structure effect. The demand for fossil energy was still larger and the energy use efficiency was lower, which led to the carbon intensity continuously increasing with the development of the economy [24]. 

The total energy consumption is the main source of regional carbon emission growth. In recent years, with the upgrade of the energy structure, the consumption of fossil fuels shows a downward trend. However, due to the fact that Jiangsu is still in the stage of industrial development, raw coal, coke and other fossil fuel make up a high proportion of the consumption in the process of industrial development; in particular, Xuzhou has long been an important energy base in Jiangsu Province, where the raw coal consumption is the highest in the province. With new, clean energy being rapidly developed, the proportion of coal in Jiangsu′s energy consumption has decreased, but it is still at a high level. The driving force of energy consumption intensity on carbon intensity is still at a high level. At the initial stage of Internet development, it brought about an increase in the carbon intensity of the region, i.e., the Internet drove the economic development of the region, which increased the energy consumption of the region and inevitably raised the carbon emissions of the region [25]. The reason for this phenomenon may be that there was an increase in electricity consumption and energy consumption due to the large investment in Internet equipment construction and the rapid increase in Internet penetration at the beginning of the Internet development, which drove the increase in carbon emissions [26]. Some studies showed that with the development of the Internet, there was a significant increase in regional technological innovation and energy efficiency, leading to increased efficiency in the operation of various industries, thus curbing the carbon intensity while the economy grew [27]. According to the current results, although the Internet development had an inhibitory effect on the carbon intensity in northern Jiangsu, it also showed an obvious promotion effect on the carbon intensity in Jiangsu, which indicated that the effect of Internet development on carbon intensity in Jiangsu has not yet reached a turning point.

## 5. Carbon Reduction Projections

The carbon emissions intensity can be used as an important indicator to judge the carbon emissions reduction potential, and the change in carbon emission intensity can reflect the carbon reduction potential. However, the pressure to reduce emissions will be greater for regions with a lower base-period carbon emissions intensity compared with those with a higher base-period carbon emissions intensity because a lower base-period carbon emissions intensity implies a higher marginal cost to further increase the share of the clean energy consumption or reduce the energy intensity, which leads to technical restrictions on related carbon emission measures. Therefore, when analyzing the carbon reduction potential, it is important to study the carbon emissions reduction potential of each region on the basis of re-comparing the carbon emission intensity of each region instead of just analyzing and comparing the changes in carbon intensity [28]. Assuming that when t = 0, the average carbon intensity is I_0_, and the carbon emission intensity changes to ∆I_t_ in year t, then the average rate of change of the carbon intensity is
(7)$$r=\Delta I_{t}{/I}_{0}$$

Assuming that the carbon intensity of each region changes according to the average change rate of Jiangsu Province, the change in the reference carbon emission intensity of region m is
(8)$$\Delta I{\left(R\right)}_{m,t}{=r*I}_{m,0}$$

In Equation (8), I_m,0_ is the carbon intensity of the base period in region m. The ratio of the actual change in carbon emission intensity to the reference change is defined as the carbon emission potential index:(9)$$I{\left(P\right)}_{m}=\Delta I_{m,t}/\Delta I{\left(R\right)}_{m,t\;}$$

In Formula (9), I(P)_m_ is the carbon emissions potential index of region m. When I (P)_m_ < 1, the actual change in carbon intensity is smaller than the reference change in carbon emissions intensity, which reflects the fact that the efficiency of carbon intensity in region m is smaller than the average value. In contrast, if I (P)_m_ > 1, it indicates that the efficiency of the carbon intensity in region m is higher than the average level.

Based on Equations (7)–(9), the carbon emissions potential indices of Jiangsu and each region were calculated using 2007 and 2018 as the base periods. The results are shown in Table 9. From the perspective of Jiangsu Province, the carbon emissions reduction potential index from 2007 to 2020 was greater than 1, while the carbon emissions reduction potential index with 2017 as the base period was less than 1. This indicated that since the 13th Five-Year Plan, Jiangsu accomplished breakthroughs in the economy but was negligent in controlling carbon emissions, resulting in a low carbon emissions reduction potential index, and faces greater pressure in energy conservation and emissions reduction, but also has greater emissions reduction potential. The index was less than 1 in both northern Jiangsu and central Jiangsu during 2007–2020 and 2016–2020, indicating that the efficiency of energy conservation and emissions reduction in the north and the center had not reached the average level of Jiangsu, and has great potential in the future. The main reason for this was that coal consumption in northern Jiangsu and central Jiangsu accounted for a relatively large proportion of the total, thus leading to an increase in carbon emissions. The carbon reduction potential index in southern Jiangsu was less than 1 during 2007–2020 and greater than 1 during 2016–2020, which indicated that the energy efficiency saving in southern Jiangsu was higher than the average value in Jiangsu during the 13th Five-Year Plan period by controlling the carbon emissions intensity while promoting economic development. Based on the good urban ecological environment in southern Jiangsu, the future emissions reduction potential is promising.

In summary, Jiangsu achieved significant progress regarding carbon reduction. However, northern Jiangsu and central Jiangsu should improve the efficiency of energy conservation and emissions reduction as soon as possible, control the consumption of high-pollution energy and improve the energy structure to reduce carbon emissions. The southern Jiangsu region should further develop science and technology while maintaining the existing level [29,30].

## 6. Conclusions and Recommendations

This study took Jiangsu Province as a sample to empirically analyze the impact of Internet development level and technological progress, economic development level, energy consumption intensity, urbanization level, energy structure and industrial scale on carbon emissions intensity. The following conclusions were found: (1) The internet development level had a significant promotion effect on the carbon emissions intensity in Jiangsu Province, but it had a trend of inhibition in southern Jiangsu. (2) The driving effect of energy consumption intensity and economic development level on the carbon emissions intensity was still significant, and the driving effect on the carbon intensity in northern Jiangsu was obviously higher than that in central and southern Jiangsu. (3) Technological progress, urbanization level and industrial scale significantly inhibited the carbon emissions intensity in Jiangsu Province. Although technological progress can reduce carbon emissions, the emissions reduction effect was less than the driving effect brought about by the development of the Internet. Therefore, the development of the Internet in Jiangsu Province needs to cross the threshold as soon as possible. This study also predicted and analyzed the future emissions reduction potential of Jiangsu Province, and found that there was still great potential for energy conservation and emissions reduction efficiency in Jiangsu Province, especially in northern and central Jiangsu regions, which should improve energy conservation and emissions reduction efficiency and control carbon emissions as soon as possible. Based on the above results, we put forward the following suggestions.

Improve energy efficiency and reduce energy intensity [31,32]. The key to reducing energy intensity is to improve energy use efficiency. Industrialization is an inevitable stage of economic development [33]. Therefore, it is necessary to speed up the breakthrough of major key technologies of negative carbon and invest corresponding research funds into key areas, such as renewable energy; zero-carbon industrial process reengineering; zero-carbon construction; carbon capture, utilization and storage, in order to improve the utilization rate of clean energy [34,35]. 

Comprehensively build a low-carbon and safe energy utilization system. Coal consumption was strictly controlled and gradually reduced as an energy source on the basis of the safety and reliability of new energy; this decrease in fossil energy consumption and increase in the proportion of renewable energy consumption should be continued [36]. At the same time, continue the strategy of traditional energy peaking and underwriting to ensure supply, and strive to improve the level of efficient utilization of coal, focusing on promoting energy-saving and consumption-reduction transformation, heat supply transformation and flexibility transformation of coal-fired power generating units [37,38].

Promote the balance between carbon emissions and economic development to accelerate low-carbon economic development [39,40]. The positive impact of economic development level on carbon emissions was significant, and although the coefficient was relatively small, the potential impact of economic development on carbon emissions is not negligible. In recent years, Jiangsu had a good economic development landscape, but the energy efficiency was relatively low; behind the rapid economic development was higher energy consumption, which also meant higher carbon emissions. Therefore, Jiangsu and all regions in the province need to find a balance between economic development and carbon emissions and develop the economy scientifically and effectively while controlling carbon emissions in order to promote low-carbon economic growth. 

According to the actual development situation in the region, the Internet should be developed according to local conditions to effectively increase the Internet penetration rate [41]. For regions with a low level of Internet development, Internet penetration is at a preliminary stage. At this time, too aggressive development of the Internet may cause an increase in carbon emissions; therefore, we should gradually promote the development of the Internet in backward regions to make them cross the threshold [42]. For Internet-developed regions to make good use of the Internet in energy conservation and emissions reduction, improve the energy use efficiency and reduce carbon emissions [43]. At the same time, we can also use the Internet platform to advocate for a low-carbon life, use the Internet social platform to promote green technology products, popularize the knowledge of low-carbon and energy conservation, drive the whole population to participate in environmental protection actions, encourage the public to participate in environmental governance together, and accelerate the formation of a green lifestyle and methods of production [44].

## Figures and Tables

**Figure 1 ijerph-19-16681-f001:**
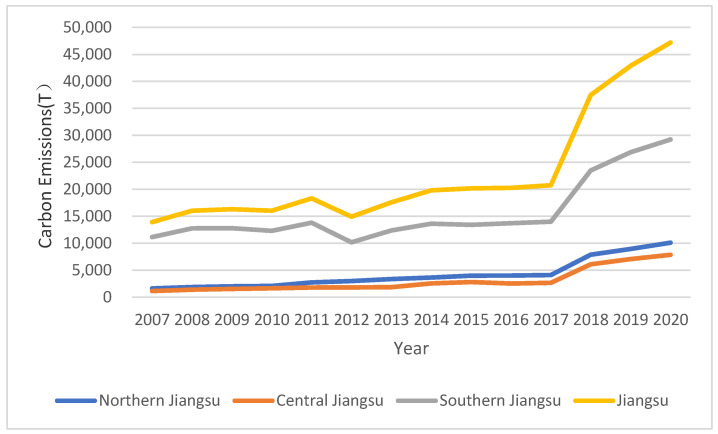
Carbon emissions in Jiangsu from 2007 to 2020.

**Figure 2 ijerph-19-16681-f002:**
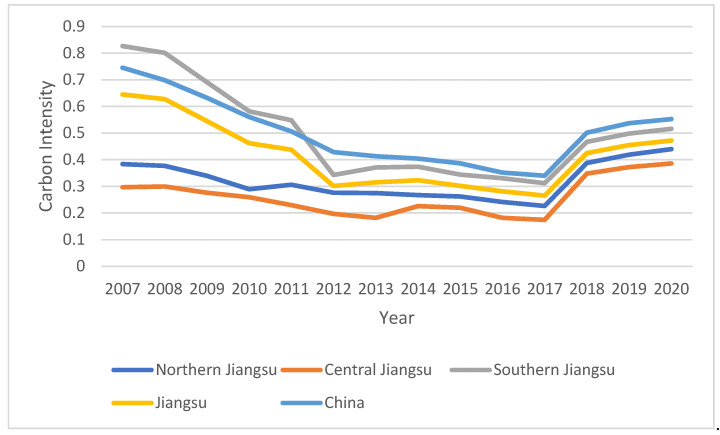
Carbon intensity in Jiangsu from 2007 to 2020.

**Table 1 ijerph-19-16681-t001:** Carbon emissions factors for each type of energy.

Types of Energy	Average Net Calorific Value	Conversion Coefficient of Standard Coal	Carbon Content Per Unit Calorific Value (t/TJ)	Carbon Oxidation Rate	Carbon Emission Coefficient
Raw Coal	20,908 kJ/kg	0.7143 kgce/kg	26.37	0.94	1.9003 kg-CO_2_/kg
Coke	28,435 kJ/kg	0.9714 kgce/kg	29.5	0.93	2.8604 kg-CO_2_/kg
Crude Oil	41,816 kJ/kg	1.4286 kgce/kg	20.1	0.98	3.0202 kg-CO_2_/kg
Fuel Oil	41,816 kJ/kg	1.4286 kgce/kg	21.1	0.98	3.1705 kg-CO_2_/kg
Gasoline	43,070 kJ/kg	1.4714 kgce/kg	18.9	0.98	2.9251 kg-CO_2_/kg
Kerosene	43,070 kJ/kg	1.4714 kgce/kg	19.5	0.98	3.0179 kg-CO_2_/kg
Diesel	42,652 kJ/kg	1.4571 kgce/kg	20.2	0.98	3.0959 kg-CO_2_/kg
LPG	50,179 kJ/kg	1.7143 kgce/kg	17.2	0.98	3.1013 kg-CO_2_/kg

**Table 2 ijerph-19-16681-t002:** Carbon emission impact factor variables.

Variables	Indicators and Units	Indicator Description	Symbols
Industrial scale	Total industrial output value above the scale	Total industrial output value above the scale at the end of each year	P
Economic development level	GDP per capital	GDP/year-end population	A
Energy consumption intensity	Total primary energy consumed per CNY 10,000 of output value	Total energy consumption/GDP	E
Urbanization level	Urbanization rate (%)	Number of urban population/total population at the end of the year	U
Energy structure	Share of raw coal consumption (%)	Share of raw coal consumption in total energy consumption	S
Technological progress	Science and technology expenditure per capita	Total amount spent on science and technology/year-end population	T
Internet development level	Internet broadband connections per capita	Total number of Internet broadband connections/total population at the end of the year	N

**Table 3 ijerph-19-16681-t003:** Descriptive statistics of the variables.

Variables	Average Value	Standard Deviation	Minimum Value	Maximum Value
lnI	−1.061	0.506	−2.299	0.197
lnP	17.95	0.937	14.88	20.41
lnA	11.03	0.72	9.062	12.2
lnE	−2.818	0.569	−3.905	−0.209
lnU	4.099	0.19	3.478	4.421
lnS	−0.686	0.449	−2.856	−0.082
lnT	5.013	1.514	−1.113	7.829
lnN	−1.802	0.905	−4.806	−0.0807
lnT*lnN	−7.843	3.134	−12.85	5.349

**Table 4 ijerph-19-16681-t004:** The results of the LLC test.

Variables	Statistic	*p*-Value
lnI	−2.4762	0.0066
lnP	−8.1222	0.0000
lnA	−6.5191	0.0000
lnE	−3.7333	0.0001
lnU	−4.6948	0.0000
lnS	−5.0954	0.0000
lnT	−4.2729	0.0000
lnN	−5.419	0.0000

**Table 5 ijerph-19-16681-t005:** The results of the Hausman test.

Variables	Coefficients		(b-B)Difference	Sqrt(Diag(V_b-V_B)) S.E.
	fe	re		
lnP	−0.2486	−0.0473	−0.2012	0.0554
lnA	0.0833	−0.0015	0.0848	0.0560
lnE	0.8037	0.8274	−0.0237	0.0342
lnU	−0.5280	−0.9952	0.4672	0.2767
lnS	0.1134	0.1026	0.0108	0.0630
lnT	−0.1170	−0.1408	0.0238	0.0082
lnN	0.4949	0.5085	−0.0136	0.0262
Constant	8.4730	7.9156	0.5574	0.8598

**Table 6 ijerph-19-16681-t006:** Regression results of the fixed-effects model.

Variable	Jiangsu		Northern Jiangsu		Central Jiangsu		Southern Jiangsu	
	Ⅰ	Ⅱ	Ⅰ	Ⅱ	Ⅰ	Ⅱ	Ⅰ	Ⅱ
lnP	−0.249 **	−0.253 **	−0.367 ***	−0.370 ***	−0.063	−0.088	0.179	0.288 *
	(−2.73)	(−2.92)	(−5.70)	(−5.66)	(−0.26)	(−0.37)	(−1.12)	(−1.92)
lnA	0.083	0.068	0.637 ***	0.678 ***	0.111	−0.088	−0.139	−0.012
	(−1.00)	(−0.83)	(−3.8)	(−3.38)	(−0.54)	(−0.37)	(−0.71)	(−0.06)
lnE	0.804 ***	0.812 ***	1.139 ***	1.141 ***	0.970 ***	0.975 ***	0.768 ***	0.836 ***
	(−9.03)	(−8.87)	(−11.67)	(−11.59)	(−3.31)	(−3.41)	(−11.25)	(−12.75)
lnU	−0.528	−0.339	−1.119 **	−1.181 **	−3.196	−2.595	−2.606 **	−3.951 ***
	(−0.88)	(−0.62)	(−2.19)	(−2.19)	(−1.59)	(−1.55)	(−2.15)	(−3.36)
lnS	0.113	0.125	−0.049	−0.052	−0.159	0.03	0.184	0.288 **
	(−1.08)	(−1.25)	(−0.49)	(−0.51)	(−0.52)	(−0.09)	(−1.5)	(−2.48)
lnT	−0.117 ***	−0.086 *	−0.074 ***	−0.098	−0.176 **	0.027	−0.185 ***	0.108
	(−3.33)	(−1.85)	(−3.28)	(−1.50)	(−2.52)	(−0.13)	(−4.58)	(−1.17)
lnN	0.495 ***	0.426 ***	0.234 ***	0.250 ***	0.838 ***	0.495	0.622 ***	−0.238
	(−4.26)	(−3.91)	(−3.19)	(−2.93)	(−4.26)	(−1.39)	(−4.71)	(−0.86)
lnT*lnN		0.01		−0.006		0.06		0.124 ***
		(−0.86)		(−0.38)		(−1.13)		(−3.47)
Constant	8.473 ***	7.768 ***	7.207 ***	7.158 ***	16.962 **	15.207 ***	12.490 ***	13.038 ***
	(−3.74)	(−3.72)	(−5.25)	(−5.15)	(−2.7)	(−2.77)	(−2.74)	(−3.11)
F test	0	0	0	0	0	0	0	0
r^2^	0.719	0.71	0.838	0.835	0.404	0.421	0.782	0.817

Notes: ***, ** and *, denote significance at the 1%, 5% and 10% levels, respectively.

**Table 7 ijerph-19-16681-t007:** Test results of the threshold effect.

Region	Threshold Variables	Threshold Model	F	P	10%	5%	1%	Threshold	95%
Northern Jiangsu	lnN	Single threshold	125.02	0	19.44	22.0191	26.2548	−1.536	[−1.5380, −1.4588]
		Double threshold	−13.53	1	14.2826	22.2827	36.1037		
Central Jiangsu	lnN	Single threshold	56.57	0.04	12.6652	14.6473	15.3348	−1.1455	[−1.2249, −1.1079]
		Double threshold	4.16	0.893	12.2012	14.8529	23.5167		
Southern Jiangsu	lnN	Single threshold	16.66	0.18	21.8862	25.4085	32.4588		
		Double threshold	16.65	0.12	17.4616	22.6775	32.212		

**Table 8 ijerph-19-16681-t008:** Regression results of the threshold effect.

Variable	Northern Jiangsu	Central Jiangsu	Southern Jiangsu
lnP	−0.268 ***	0.209	0.244 *
	(−5.92)	(−1.48)	(1.75)
lnA	0.285 **	0.047	0.024
	(−2.35)	(−0.41)	(0.15)
lnU	−0.35	−3.902 ***	−3.315 ***
	(−1.00)	(−3.46)	(−3.29)
lnE	1.121 ***	0.750 ***	0.789 ***
	(−17.39)	(−4.56)	(13.95)
lnS	−0.1	−0.008	0.287 ***
	(−1.52)	(−0.05)	(2.77)
lnT	0.004	−0.075 *	−0.163 ***
	(−0.25)	(−1.71)	(−4.22)
lnN ≤ −1.5360	−0.004		
	(−0.08)		
lnN > −1.5360	−0.209 **		
	(−2.42)		
lnN ≤ −1.1415		0.438 ***	
		(3.20)	
lnN > −1.1415		−0.217	
		(−1.01)	
Constant	4.951 ***	13.651 ***	12.719 ***
	(−6.27)	(3.84)	(3.32)
F test	0	0	0
r2	0.932	0.817	0.852

Notes: ***, ** and * denote significance at the 1%, 5% and 10% levels, respectively.

**Table 9 ijerph-19-16681-t009:** Change of carbon emissions intensity and potential index in Jiangsu.

Region	2007–2020			2016–2020		
	Reference Change (t/CNY 10,000)	Actual Change (t/CNY 10,000)	Potential Index	Reference Change (t/CNY 10,000)	Actual Change (t/CNY 10,000)	Potential Index
Northern Jiangsu	0.0576	0.0566	0.9824	0.2220	0.2137	0.9625
Central Jiangsu	0.1012	0.0892	0.8816	0.2129	0.2118	0.9946
Southern Jiangsu	−0.3491	−0.3106	0.8896	0.1954	0.2039	1.0436
Jiangsu	−0.1138	−0.1728	1.5179	0.2247	0.2069	0.9205

## Data Availability

The datasets used and/or analyzed during the current study available from the corresponding author on request.

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
