# Peer review of "Impact of Internet Development on Carbon Emissions in Jiangsu, China"

_ijerph, 2022, doi:10.3390/ijerph192416681_

Round 1
Reviewer 1 Report
Dear authors,
your research is very interesting and current. In this sense, I have some major recommendations in order to improve your manuscript.
1. Abstract: the first sentence of the abstract must be reformulated (it is too long and loses its meaning). The abstract must contain short and concise sentences regarding the purpose of the work, the methodology used and the results obtained. I recommend reviewing the abstract.
2. Introduction: in this section you should better justify the necessity and novelty of your research. To what extent does this research cover the existing gaps in the field? What are the targets pursued at the country level in order to develop the Internet and reduce carbon emissions?
3. Research Design:
- Table 1 should be redone. It is inserted in the text as a picture and it is not clearly visible.
- line 143-144: In the choice of analysis methods, most of them use .... some works that used these methods should be cited
- the abbreviations used in the text must be explained, for example Impact, Population, Affluence, Technology (IPAT)
- line 154: t should be T?
- are you sure that in equations 3, 4, 5 and 6 the free term a (model coefficient) and the standard error ε must also be logarithmized?
- Table 2 should also be redone. It is inserted in the text as a picture and it is not clearly visible.
4. Results
- Sections 3.3., 3.4 and 3.5 should be grouped in a separate section tentatively named 4. Results and discussions
- I consider it necessary to insert some tables with the results of the unit root test and one with the results of the Hausman test
- An explanation of the symbol *, **, *** must be added as a note under the table 4
- Tables 4 and 5 should also be redone. It is inserted in the text as a picture and it is not clearly visible.
5. Conclusions
- Sections 4 and 5 should be compressed in one section. The conclusions should be better formulated and the emphasis should be placed on the results obtained compared to other studies.
Author Response
The author wishes to express sincere gratitude to the Editor and Reviewers for these useful comments. According to these comments, we give a careful answer and revision. We hope that our revisions are sufficient enough to convince the Editor and Reviewers that this research is a significant work and conducted with great efforts.

Reviewer 2 Report
This paper reports a thorough investigation into impact of internet development on carbon emissions in Jiangsu province. The paper provides some useful findings and observations, although some comments should be addressed as listed below.
1. The latest research studies on STIRPAT model should be included in the literature review, for example:
Yu, S., Zhang, Q., Hao, J. L., Ma, W., Sun, Y., Wang, X., & Song, Y. (2023). Development of an extended STIRPAT model to assess the driving factors of household carbon dioxide emissions in China. Journal of Environmental Management, 325, 116502.
2. Tables 2, 4 & 5 should be reproduced in high resolution.
3. Why did the authors choose Jiangsu Province as a specific province to study? Is it representative?
4. The conclusion section should be expanded. The current conclusions are rather weak.
Author Response

(The authors gave the same response as above.)

Reviewer 3 Report
This manuscript analyzes the effects of economic level, industrial scale, energy structure and technological progress on carbon emissions in Jiangsu Province under the background of Internet development, and predicts future emission reduction potential, which is of great significance for the realization of carbon emission reduction and low-carbon Internet development. It is an interesting paper. The topic is important and appropriate for IJERPH. This manuscript can be accepted provided that the following concerns are answered or resolved.
(1)Abstract, authors are suggested to start broad in the general background, then narrow in on the relevant topic that will be pursued in the paper. Maybe this part can be improved.
(2)This paper does not shows an adequate review of the relevant literature and cite accordingly. Many of the recent works have been ignored, such as Interaction between Digital Economy and Environmental Pollution: New Evidence from a Spatial Perspective; Coordinating socio-economic and environmental dimensions to evaluate regional sustainability -towards an integrative framework; Cross-Regional Comparative Study on Environmental–Economic Efficiency and Driving Forces behind Efficiency Improvement in China: A Multistage Perspective; Has the Digital Economy Reduced Carbon Emissions?: Analysis Based on Panel Data of 278 Cities in China; High energy-consuming industrial transfers and environmental pollution in China: A perspective based on environmental regulation and so on.
(3)The authors must review his works through an appropriate translation review service, paying particular attention to English grammar, spelling, and sentence structure , so that the goals and results of the study are clear to the reader.
(4)The list of references is not in our style. It is close but not completely correct. Please refer to the format instructions for the author.
Author Response

(The authors gave the same response as above.)

Round 2
Reviewer 1 Report
Dear authors, I appreciate that you took into account all the observations made and responded accordingly. Congratulations for the research undertaken.
Reviewer 2 Report
All the comments have been well addressed by the authors in the revised manuscript.